# Surgical Outcome of Children with a Malignant Liver Tumour in The Netherlands: A Retrospective Consecutive Cohort Study

**DOI:** 10.3390/children9040525

**Published:** 2022-04-07

**Authors:** Merel B. Klunder, Janneke L. M. Bruggink, Leon D. H. Huynh, Frank A. J. A. Bodewes, Alida F. W. van der Steeg, Kathelijne C. J. M. Kraal, C. P. (Kees) van de Ven, Martine van Grotel, József Zsiros, Marc H. W. A. Wijnen, I. Q. (Quintus) Molenaar, Robert J. Porte, Vincent E. de Meijer, Ruben H. de Kleine

**Affiliations:** 1Department of Surgery, Division of Hepato-Pancreatico-Biliary Surgery and Liver Transplantation, University of Groningen, University Medical Center Groningen, 9713 GZ Groningen, The Netherlands; m.b.klunder@umcg.nl (M.B.K.); r.j.porte@umcg.nl (R.J.P.); v.e.de.meijer@umcg.nl (V.E.d.M.); 2Department of Surgery, Division of Pediatric Surgery, University of Groningen, University Medical Center Groningen, 9713 GZ Groningen, The Netherlands; j.l.m.bruggink@umcg.nl; 3Department of Surgery, Princess Máxima Center for Pediatric Oncology, 2584 CS Utrecht, The Netherlands; l.d.h.huynh@students.uu.nl (L.D.H.H.); a.f.w.vandersteeg@prinsesmaximacentrum.nl (A.F.W.v.d.S.); c.p.vandeven-4@prinsesmaximacentrum.nl (C.P.v.d.V.); m.h.w.wijnen-5@prinsesmaximacentrum.nl (M.H.W.A.W.); 4Department of Pediatric Hepatology and Gastroenterology, University of Groningen, University Medical Center Groningen, 9713 GZ Groningen, The Netherlands; f.a.j.a.bodewes@umcg.nl; 5Department of Pediatric Oncology, Princess Máxima Center for Pediatric Oncology, 2584 CS Utrecht, The Netherlands; k.c.j.kraal@prinsesmaximacentrum.nl (K.C.J.M.K.); m.vangrotel@prinsesmaximacentrum.nl (M.v.G.); j.zsiros@prinsesmaximacentrum.nl (J.Z.); 6Department of Surgery, University of Utrecht, University Medical Center Utrecht, 2584 CX Utrecht, The Netherlands; i.q.molenaar@umcutrecht.nl

**Keywords:** liver, cancer, paediatric, surgery

## Abstract

Introduction: Six to eight children are diagnosed with a malignant liver tumour yearly in the Netherlands. The majority of these tumours are hepatoblastoma (HB) and hepatocellular carcinoma (HCC), for which radical resection, often in combination with chemotherapy, is the only curative treatment option. We investigated the surgical outcome of children with a malignant liver tumour in a consecutive cohort in the Netherlands. Methods: In this nationwide, retrospective observational study, all patients (age < 18 years) diagnosed with a malignant liver tumour, who underwent partial liver resection or orthotopic liver transplantation (OLT) between January 2014 and April 2021, were included. Children with a malignant liver tumour who were not eligible for surgery were excluded from the analysis. Data regarding tumour characteristics, diagnostics, treatment, complications and survival were collected. Outcomes included major complications (Clavien–Dindo ≥ 3a) within 90 days and disease-free survival. The results of the HB group were compared to those of a historical HB cohort. Results: Twenty-six children were analysed, of whom fourteen (54%) with HB (median age 21.5 months), ten (38%) with HCC (median age 140 months) and one with sarcoma and a CNSET. Thirteen children with HB (93%) and three children with HCC (30%) received neoadjuvant chemotherapy. Partial hepatic resection was possible in 19 patients (12 HB, 6 HCC, and 1 sarcoma), whilst 7 children required OLT (2 HB, 4 HCC, and 1 CNSET). Radical resection (R0, margin ≥ 1.0 mm) was obtained in 24 out of 26 patients, with recurrence only in the patient with CNSET. The mean follow-up was 39.7 months (HB 40 months, HCC 40 months). Major complications occurred in 9 out of 26 patients (35% in all, 4 of 14, 29% for HB). There was no 30- or 90-day mortality, with disease-free survival after surgery of 100% for HB and 80% for HCC, respectively. Results showed a tendency towards a better outcome compared to the historic cohort, but numbers were too small to reach significance. Conclusion: Survival after surgical treatment for malignant liver tumours in the Netherlands is excellent. Severe surgical complications arise in one-third of patients, but most resolve without long-term sequelae and have no impact on long-term survival.

## 1. Introduction

Liver tumours are extremely rare in children under the age of 18 years, and care for these patients is highly demanding. About two-thirds of liver tumours are malignant, and the incidence is estimated at 2.4 per 1 million per year, corresponding to about seven diagnosed paediatric liver tumours each year in the Netherlands [1,2]. Hepatoblastoma (HB) is the most frequently occurring paediatric liver tumour, accounting for two-thirds of the malignant liver tumours, followed by hepatocellular carcinoma (HCC) with around 20 percent. HB occurs mostly in the first 5 years of life, whereas HCC occurs mostly in children older than 10 years of age [3,4,5]. The third and remaining group of malignant paediatric liver tumours consists of a broad range of very rare diagnoses, including biliary rhabdomyosarcoma, malignant rhabdoid tumour, undifferentiated sarcoma and calcifying nested stromal tumours (CNSET) [6].

The survival of HB and HCC is highly dependent on the resectability of the tumour. Less than 20% of the HCC are resectable at the time of diagnosis. Since HCC does not respond well to platinum-based chemotherapy regimens, as was seen in the SIOPEL 2 and 3 studies, the road to resection may be challenging. After chemotherapy, complete resection may be achieved in about 40% of the HCCs. Five-year overall survival rates corresponding with complete surgical resection have been reported at 59%. Whereas 5-year overall survival for HCC in general is 22% [7,8]. Radical resection or thermal ablation remains the only curative treatment options in both tumour types. Pre-operative chemotherapy may be administered to treat extrahepatic and metastatic disease and reduce tumour size, rendering a patient eligible for surgery [9,10,11,12]. When partial liver resection does not completely remove the tumour, whilst preserving a sufficient liver remnant, orthotopic liver transplantation (OLT) should be considered [11,12,13,14].

Since June 2018, all children with cancer in the Netherlands are referred to the Princess Máxima Center (PMC) for Pediatric Oncology in Utrecht in a shared care construction. The surgical treatment of children with a liver disease is located at the University Medical Center Groningen (UMCG), the national centre for Hepato-Pancreatico-Biliary Surgery and Liver Transplantation in children. Surgical treatment of paediatric liver cancer patients is performed in a dedicated collaboration with the surgical department of the PMC [15].

The aim of this study was to review the clinically significant surgical complications of HB and HCC resection and transplantation within 90 days after surgery. The secondary aim was to assess disease-free survival. Additionally, we compared our results with a historical, previously published cohort of Dutch HB patients [16].

## 2. Materials and Methods

All consecutive patients with a paediatric primary liver malignancy in the Netherlands, who received surgical treatment in the period from January 2014 to April 2021, were evaluated for inclusion in a retrospective observational study. All patients had been included in a structured follow-up, yielding complete data sets without the need for extra investigations. Children with a malignant liver tumour who were not eligible for surgery were excluded from the analysis. The medical charts from both centres were examined by MK, JH, AvdS and RdK to collect information regarding demographics, diagnostics, treatment, complications and follow-up.

Approval from the medical ethics committee (METC) of the UMCG was obtained, and the need for informed consent was waived. The study was registered in the local research registry of the UMCG prior to the initiation of the study (RR 202000787, 12 January 2021). This study was performed according to the STROBE initiative [17]. A database was built in the Research Electronic Data Capture (REDCAP) system and statistical analyses were performed using SPSS (Windows version 23).

### 2.1. Statistical Analysis

Patient, tumour and treatment characteristics were described using frequency tables. For normally distributed continuing variables, mean and standard deviation were used. For not normally distributed data, median and interquartile range (IQR) were used. The Fisher’s exact test was used to compare nominal data. The Mann–Whitney U test was used to compare continuous skewed data. Disease-free survival was calculated from the date of surgery until the date of recurrence of oncologic disease related to a liver tumour or a last follow-up, using Kaplan–Meier analysis. A *p*-value of <0.05 was considered statistically significant.

### 2.2. Diagnosis and Treatment

The diagnosis of a malignant liver tumour was confirmed in a multidisciplinary team meeting and was based on laboratory markers (AFP levels at presentation) [18], radiological findings (ultrasonography of the abdomen and CT and/or MRI of the abdomen), and histological results (biopsy or resection). The PRETEXT/POSTTEXT system was used when appropriate to determine the extent of disease, evaluating tumour extension, vessel involvement and metastatic disease [19,20]. A CT scan of the thorax was used to assess the presence of pulmonary metastases in accordance with SIOPEL 4 guidelines [21]. After confirmation of the diagnosis, patients were evaluated for primary resection or the need for neoadjuvant chemotherapy following SIOPEL or the more recent PHITT-1 guidelines.

All surgical procedures in the Netherlands were performed at either the UMCG or the PMC. The type of resection was based on the tumour location and extent of disease, striving for a surgical margin of healthy tissue between tumour and the plane of resection. The date of resection, the intraoperative use of ultrasonography, the length of the procedure (in minutes), the amount of blood loss and any intraoperative complications were recorded. The resection margin was determined by macroscopic intraoperative evaluation of 1 cm slices. This was performed by a dedicated hepatobiliary pathologist, with microscopic verification in the following week. Complete tumour removal via radical resection was defined as a macroscopic margin of ≥1.0 mm from the shortest border of the tumour to the resection plane, irrespective of the extent of the area [22].

### 2.3. Postoperative Complications

A postoperative complication was defined as any deviation from the normal expected course up to the 30- and 90-day interval, including complications occurring after discharge. The Clavien–Dindo (CD) classification was used to reliably score the clinical impact of a postoperative surgical complication [23,24]. Only complications classified as CD 3a (reintervention under local anaesthesia), CD 3b (reintervention under general anaesthesia) or CD 4 (life-threatening complications requiring admittance to ICU) were analysed further. The postoperative complications were divided into the following five groups:Haemorrhage: defined as perioperative blood loss requiring reoperation or blood transfusion;Infection: any infection clinically related to the surgical procedure;Biliary complications: biliary tract stricture or bile leak;Vascular complications other than haemorrhage: compromised hepatic blood flow after surgery;Other complications: any other postoperative complication due to surgery requiring prolonged hospital stay or reintervention not included in any of the previous categories.

### 2.4. Follow-Up

Follow-up was performed on all patients according to the international treatment protocol. This included assessing AFP levels, chest X-rays and ultrasonography of the abdomen at regular intervals to monitor for any residual or recurring disease.

## 3. Results

### 3.1. Clinical Features

Between January 2014 and April 2021, 41 patients were diagnosed with a malignant liver tumour. Six patients (four HCC, one HB and one sarcoma) with HB were excluded since these patients were not eligible for resection or transplantation after an optimal chemotherapeutic regimen. Nine patients were excluded from analysis due to admittance in a worldwide trial prohibiting data analysis outside of the trial (Figure 1) [25]. The remaining 26 patients (14 HB, 10 HCC, 1 sarcoma and 1 CNSET) received surgery and were further analysed with patient and tumour characteristics depicted in Table 1. Metastatic disease at the time of diagnosis was present in six patients (6/26 patients, 4/14 HB (29%), 2/10 HCC (20%)). Five of the operated HB patients had pulmonary metastases at diagnosis and one HCC patient had positive lymph nodes located at the area of the left adrenal gland and the precardial superdiaphragmal area.

### 3.2. Treatment

Seventeen patients received neoadjuvant chemotherapy (13/14 HB, 3/10 HCC and 1 sarcoma). In the HB group, 12 of the 14 (86%) patients underwent partial liver resection and 2 (14%) patients underwent an OLT. Both of these patients were diagnosed with PRETEXT IV with multifocal lesions. The sarcoma patient underwent partial liver resection, and the CNSET patient received an OLT. In the HCC group, 6 out of 10 (60%) patients underwent partial liver resection opposed to 4 (40%) patients who underwent an OLT (Table 2). Metastatic disease found at the time of diagnosis was in all cases solely treated with chemotherapy, not requiring additional surgery. All pulmonary metastases found at presentation were non-detectable at the end of treatment. HCC, which originated in the background of liver disease, was found in four patients of whom its presence was unknown in three patients diagnosed with cirrhosis.

### 3.3. Radicality

The resection margins of the patients who underwent surgery were ≥1.0 mm in 24 (92%) patients, indicating complete tumour removal. In two patients, the distance between tumour margin and the resection plane was focally less than 1.0 mm, but no macroscopical tumour was left behind (R1 resection). These children underwent an extended right hemihepatectomy and an extended left hemihepatectomy and were diagnosed with a POSTTEXT 2 and POSTTEXT 3 HB, respectively. Both patients received adjuvant chemotherapy and have so far been free of recurrent disease after 60 and 32 months of follow-up, respectively.

### 3.4. Complications

Perioperative complications (both intraoperative and postoperative) requiring additional action during primary surgery or reintervention in the postoperative course (≤90 d) were registered in nine (35%) patients. In total, 17 perioperative complications occurred (Table 3). Major complications (CD ≥ 3a) were found in 5 out of 7 patients after OLT (71%) and 4 out of 19 patients after partial liver resection (21%).

One intraoperative complication requiring cardiopulmonary resuscitation during surgery was due to hypovolemic shock in combination with a possible air embolus. Complete recovery occurred postoperatively.

In patients without complications, intraoperative blood loss was less (median 220 mL, IQR 55–395 mL) than in patients with complications (median 1700 mL, IQR 575–3225 mL, *p* = 0.003). The median operation time was 341 min (IQR 227.5–457 mL) in uncomplicated patients as opposed to a median of 590 min (IQR 478–734.5 min) in patients with perioperative complications (*p* = 0.002).

Postoperative haemorrhage occurred in two patients with surgical removal of a haematoma of which one patient still had an active bleeding site.

Infection: Four patients developed a severe infection, septic shock requiring re-intubation, persisting bacteraemia due to a venous access port (VAP) infection, intra-abdominal abscess formation requiring percutaneous drainage or respiratory failure due to a pneumonia.

Biliary complications did not occur in either patient group.

Vascular complications other than haemorrhage scored as CD ≥ 3a did not occur in any patient. Besides these complications, 11 complications did not fit the criteria of the groups mentioned above. Acute liver graft rejection was reported once, which responded well to corticosteroid treatment. Poor primary function after OLT requiring re-OLT was reported once. One patient developed an obstructive ileus, requiring additional surgery due to intra-abdominal adhesions. Three patients with complications required admittance to the ICU, one due to acute kidney failure, one due to circulatory insufficiency and one due to increased abdominal pressure after surgery requiring sedation and mechanical ventilation. Furthermore, one patient suffered from persistent ascites production and another patient demonstrated pleural effusion requiring drainage under sedation. Lastly, one patient had continuous abdominal chylous production requiring the re-insertion of a PICC line under sedation for administering parenteral nutrition. The severity of the complications according to the CD classification are depicted in Table 3.

### 3.5. Survival of Operated Patients

The median follow-up time was 39.6 months for the study group and 40.0 months for HB and HCC, respectively. Ninety-day survival after surgery was 100% for the entire group. In the HB group, all patients were alive the last time checked in the follow-up. Therefore, the disease-free and overall survival was 100%. Two patients in the HCC group died due to progressive metastatic disease after 34 and 41 months, resulting in a disease-free survival of 80%. Their tumours were a PRETEXT III with V+ and P+ and PRETEXT II with E+ and N+ classification with positive lymph nodes at the area of the left adrenal gland and the precardial superdiaphragmal area. Both patients underwent radical resection with free margins without liver recurrence. There were no deaths due to other causes, making the overall survival identical. In Figure 2, the disease-free survival is shown for the HB and HCC patients, respectively. Kaplan–Meier survival analysis was used. The CNSET patient showed recurrence during follow-up and is alive with disease. The patient with sarcoma is free of disease and doing well.

### 3.6. Comparison to the Historical Group

We compared the study data of the 14 operated HB patients with a historical surgical group, with data available on 73 of the 76 resected patients (Table 4). Similar to our cohort, no children were included in this previous study who were not eligible for surgery. In the historical group, the complication rate of the transplanted patients was not included in the publication; therefore, only a comparison with the partial liver resection patients could be made. With a *p*-value of 0.0774, a trend towards fewer complications can be seen in the study group. Bleeding was found in the study group, with haemorrhage occurring in 1 out of 14 vs. 33 out of 73 historical patients. Relapse of disease in HB patients was found in 0/14 vs. 17/73 patients. Finally, HB prognosis was 100% disease-free survival after a 40-month follow-up in the study group vs. 77% after a 5-year follow-up in the historical HB resection group (*p* = 0.1199).

## 4. Discussion

### 4.1. Complications

This study shows that complications after liver surgery for malignant liver tumours are observed in one-third of patients.

Both intraoperative blood loss and duration of the surgical procedure are correlated to the occurrence of perioperative complications, and we believe this inherent risk is related to the extent of the resection. However, we could not demonstrate a direct correlation between the occurrence of complications and the extent of the surgery. Furthermore, no significant correlation was found between the POSTTEXT staging and the occurrence of perioperative complications.

In the HCC group, 40% of the patients underwent an OLT as opposed to 14% of patients in the HB group. The higher number of OLTs performed in the HCC group can be attributed to the fact that three of these patients underwent an OLT due to an underlying liver condition with cirrhosis. None of the transplanted patients showed recurrence of HCC. In our study, we found a high percentage of major liver resections that, together with the procedure of a liver transplant, can lead to a high rate of complications. The decision for the optimal surgical strategy, between resection, transplantation or additional chemotherapy, requires an experienced multidisciplinary team, preferably via a centralised national approach.

The relatively high rate of grade CD 3b complications is partly explained by the fact that general anaesthesia during an intervention in children is frequently warranted, whereas in adults, local anaesthesia is often considered sufficient. In other international studies, complication rates were reported between 15.5% and 69% [26,27,28]. Tannuri et al. showed a low postoperative complication rate; however, the reported postoperative complications seem to only have a direct connection to the operation performed, whereas in our study, all complications during postoperative admission were gathered.

### 4.2. Mortality

This study shows a 90-day survival of 100% and 40-month disease-free survival of 100% and 80% for hepatoblastoma and hepatocellular carcinoma, respectively. The prognosis of the patient seems directly related to tumour biology and not to treatment-related complications. Low mortality rates due to postoperative complications are also reported in the other studies published by specialised centres [26,27].

### 4.3. Historical Controls

We compared the rate of complications with a historical group of resected HB patients to a study group of resected and transplanted HB patients. This could have led to an underestimation of the total amount of complications in the historical group, explaining why we did not find a clear clinically significant difference but a trend between the results of the two small groups (*p* = 0.0774).

Complete tumour removal with resection margins ≥ 1.0 mm was achieved in all but two HB patients (92%). This data is not available for the historical cohort, but up to 17 patients have tumour recurrence, with the liver being affected four times [16]. Due to the limited numbers, we could not find a clear improvement in the rate of radical resection when comparing the two cohorts (*p* = 0.119), but we believe a recurrence in the liver can negatively influence prognosis.

### 4.4. Patients, Inclusion and Exclusion

Only patients who were surgically treated were included in the study. Several patients had to be excluded due to legal restrictions resulting from their participation in a different study. Survival analysis in this study is thus only applicable to HB and HCC patients who received surgical treatment and does not give a reliable insight into overall survival for all HB and HCC patients, especially since the patients who were not eligible for surgery are known to have a worse prognosis. The disease-free survival for HB and HCC patients receiving surgical treatment in this study was found to be 100% and 80%, respectively. We are confident that a statistically significant difference will be found showing a clear improvement in outcomes, including survival, once the data of the other operated patients become available. Since the formal centralisation started in the Netherlands in 2018, patient numbers were too limited to statistically perform a study into the effect of this nationalised care.

### 4.5. Incidence

An estimated 3.3 million children were living in the Netherlands, yielding an incidence of 0.91 per million per year for HB and 0.65 for HCC. This incidence is similar to the historical cohort.

### 4.6. Strengths and Limitations

The retrospective design of the study, over a time period of 7 years, created slight differences in diagnostic and treatment approaches; SIOPEL 4 protocols were used in the beginning. PHITT protocols are used currently. In several cases, the PRETEXT/POSTTEXT classification was determined in retrospect by either the clear description of tumour location and tumour extension in the imaging report or by a review of the scans by a hepatobiliary surgeon specialised in paediatric liver surgery in order to apply a corresponding PRETEXT/POSTTEXT classification. Currently, all scan reports mention a PRETEXT or POSTTEXT staging. The number of patients is limited. This is mostly due to the fact that paediatric liver tumours are extremely rare. A longer time period could allow for a higher number of patients in the study groups. A larger study group, such as the PHITT group, could give a better insight into the outcome of hepatoblastoma and HCC after the centralisation of paediatric oncology care in Europe. A worldwide database would be the preferred method to conduct a study on the outcome for HB and HCC as this ensures higher patient numbers. The CHIC database presents a good step towards the creation of a worldwide database, as the four biggest research groups in the world focusing on paediatric liver tumours (SIOPEL, COG, JPLT, Gesellschaft für pädiatrische Onkologie und Hämatologie (GPOH)) have contributed their data to this database [29].

## 5. Conclusions

Survival after surgical treatment for malignant liver tumours in the Netherlands is excellent. Severe surgical complications arise in one-third of patients, yet most resolve without long-term sequelae and have no impact on long-term survival. This rate of complications mirrors the severity of the disease and the extent of the surgical procedures. Since centralisation of this type of paediatric surgery in the Netherlands, a trend towards lower complications rates and improved survival has been observed.

## Figures and Tables

**Figure 1 children-09-00525-f001:**
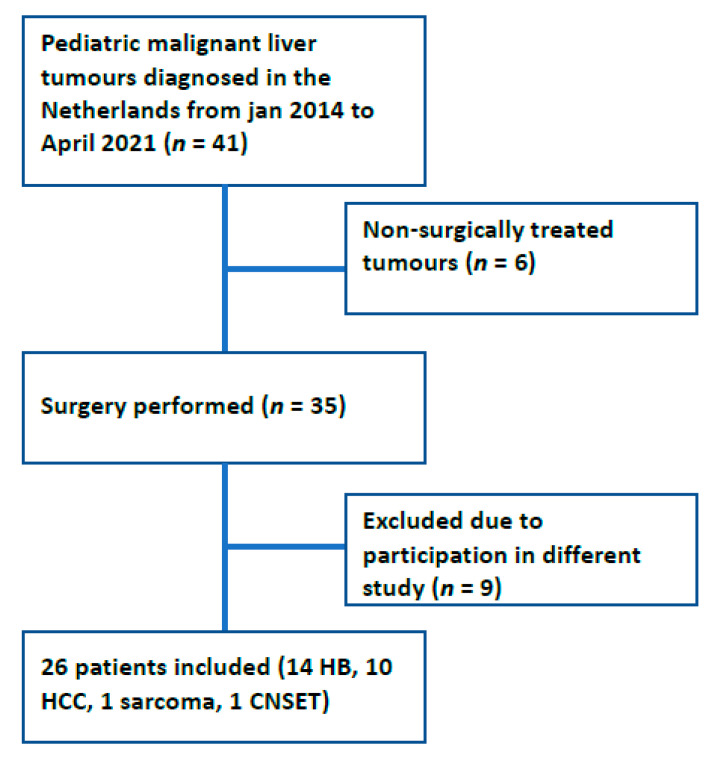
Flow chart of inclusion and exclusion. HB: hepatoblastoma; HCC: hepatocellular carcinoma; CNSET: calcifying nested stromal epithelial tumour.

**Figure 2 children-09-00525-f002:**
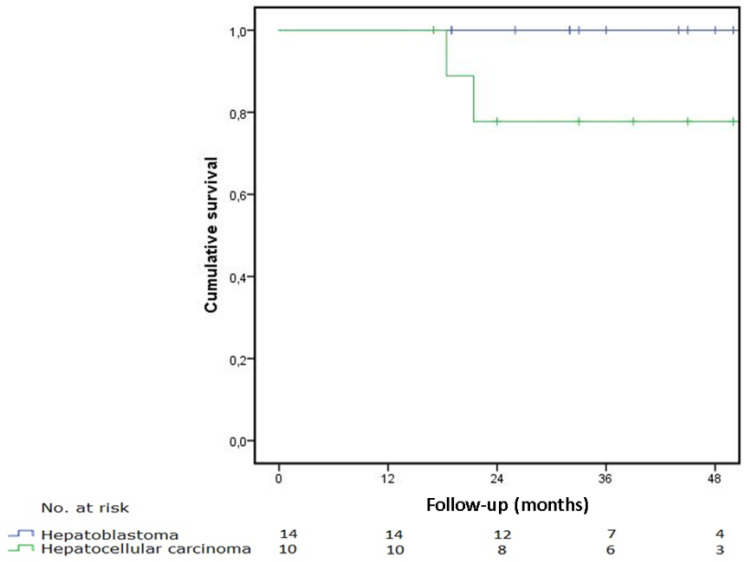
Disease-free survival of HB and HCC patients.

**Table 1 children-09-00525-t001:** Patient characteristics.

Category	*Number*	(*%*)
Sex		
Male	14	54
Female	12	46
Age (months)		
Median	49	
Interquartile range	17.5–153	
AFP (ug/L)		
Mean	180,367	
Median	2300	
Interquartile range	15.1–275,000	
PRETEXT		
I	2	8
II	8	31
III	8	31
IV	3	11
Not applicable	5	19
Hepatoblastoma	14	54
Mixed epithelial/mesenchymal	11	79
**Epithelial**	3	21
Hepatocellular carcinoma	10	38
Sarcoma	1	4
Calcifying Nested Stromal Epithelial Tumour	1	4

**Table 2 children-09-00525-t002:** Overview of surgical procedures for paediatric liver tumours.

	All Liver Tumours		HB		HCC	
Type of Surgery	*Number*	(*%*)	*Number*	(*%*)	*Number*	(*%*)
Partial liver resection	19	73	12	86	6	60
Segmentectomy/local excision	5	19	1	7	3	30
Hemihepatectomy left	3	12	2	14	1	10
Hemihepatectomy right	4	15	3	21	1	10
Extended hemihepatectomy left	3	12	3	21	0	0
Extended hemihepatectomy right	4	15	3	21	1	10
Total hepatectomy with orthotopic liver transplantation	7	27	2	14	4	40

**Table 3 children-09-00525-t003:** Distribution of complications by severity (CD classification).

Category	Partial Liver Resection *n* = 19	OLT *n* = 7
Haemorrhage	1 (1 ^b^)	2 (2 ^a^)
Clavien–Dindo 3b	0	2
Clavien–Dindo 4b	1	0
Infection	3	1 (1 ^a^)
Clavien–Dindo 3a	1	0
Clavien–Dindo 3b	0	1
Clavien–Dindo 4a	2	0
Biliary	0	0
Vascular	0	0
Other	2	9 (3 ^a^)
Clavien–Dindo 3a	0	1
Clavien–Dindo 3b	1	6
Clavien–Dindo 4a	1	1
Clavien–Dindo 4b	0	1

^a^ Number of patients who required additional surgery. ^b^ Intraoperative complication.

**Table 4 children-09-00525-t004:** Direct comparison between study cohort and historical cohort of HB in the Netherlands.

	All Liver Tumours *n* = 26 (%)	HB *n* = 14 (%)	HB Historical Group *n* = 94 (%)	*p* Value
Complication percentage	9 (35)	4 (29)	42/73 (54)	0.0774
Haemorrhage	3 (12)	1 (7)	33/73 (45)	0.0071 *
Biliary	0 (0)	0 (0)	9/73 (13)	0.344
Vascular	0 (0)	0 (0)	2/73 (3)	1.0
Infection	4 (15)	3 (21)	6/73 (8)	0.1548
Other	7 (27)	3 (21)	NA	
Recurrent disease	2 HCC, 1 CNSET AWD (12%)	0 (0)	Total 16/94, liver 3/94, pulmonary 12/94, both 1 (17)	0.1199
OLT	7 (27)	14 (2)	18/94 (19)	1
Survival 30 d.	100 (nr at risk 26)	100 (nr at risk 14)	100	
Survival 90 d.	100 (nr at risk 26)	100 (nr at risk 14)	NA	
Survival 1 jr.	100 (nr at risk 25)	100 (nr at risk 14)	NA	
Survival 5 jr.	92 (nr at risk 4)	100 (nr at risk 2)	82 (77/94)	0.1199

* *p* value of <0.05 is considered statistically significant in the Fisher’s exact test.

## Data Availability

The data are not publicly available due to privacy reasons.

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
