# Peer review of "Surgical Outcome of Children with a Malignant Liver Tumour in The Netherlands: A Retrospective Consecutive Cohort Study"

_children, 2022, doi:10.3390/children9040525_

Round 1

Reviewer 1 Report

Klunder et al. deal with an important and rare topic on paediatric malignant tumours. The work has a high value and it must be taken into account that we have little data on this disease in children and its surgical treatment.
Since the incidence of this disease is low, the chosen metohde of evaluation seems adequate.
The study period was chosen from 2014 to 2021, why limit the follow-up to 48 months?
Disease free survival is shown for HCC versus HB and for 48 months.  A presentation that compares the historical with the current study cohort according to the paper design and, if applicable, compares HCC with HB patients in general would be exciting.
The tables should urgently be improved in their presentation format.
While table 2 reports two HB patients on OLT, table 4 reports a total of 14 patients compared to the historical cohort.
For the comparison between liver resection and liver transplantation, the difference between the historical and the current study cohort was not taken into account.
Were there no HCC patients in the historical cohort?

Author Response

Since the incidence of this disease is low, the chosen metohde of evaluation seems adequate.The study period was chosen from 2014 to 2021, why limit the follow-up to 48 months?
Disease free survival is shown for HCC versus HB and for 48 months.  

Follow-up was collected up to 5 years or death. The KM curve was cut off at 48 months for statistical reasons because only 2 patients that were included in the study reached more than 60 months. In table 4 this can be checked. KM of up to 60 months is available but the low number at risk will show a drop below the 2 patients that actually died due to recurrence. This is a statistical disadvantage for reasons of calculated and perceived risk in the KM statistical method and might induce even more confusion.

A presentation that compares the historical with the current study cohort according to the paper design and, if applicable, compares HCC with HB patients in general would be exciting.

We fully agree with the suggestion of the reviewer. Unfortunately this study is not possible. Unfortunately the historical data in its raw form is not available for direct comparison. This was verified via correspondence with one of the senior authors. We extracted the data from the publication itself and incorporated this into the paper for comparison.

The tables should urgently be improved in their presentation format.

We agree. We hope MDPI will adept the style and layout to more suit publication.

While table 2 reports two HB patients on OLT, table 4 reports a total of 14 patients compared to the historical cohort.

In the historical cohort only resected patients are available, in the study cohort this group exists of resected and transplantated patients (14 total, 12 and 2, table 4).. 

For the comparison between liver resection and liver transplantation, the difference between the historical and the current study cohort was not taken into account.
Were there no HCC patients in the historical cohort?

There were no HCC patients included in the cohort, so a comparison could only be made on the HB patients. We made the comparison of historically resected HB vs resected and transplanted HB.

Reviewer 2 Report

Excelent article, I applaud your work, both surgical and scientific. The entirety of the article is well written, with very clear explanation to the process, results, and important discussions. Nevertheless, I would like you to recheck the data in the tables - for example in Table 1, AFP values are very high - please check.

The only drawback I can find to the article is the low number of cases. I hope that you will be able to conduct a larger study, maybe prospective, in the future. Also, I hope that the patients in the present study will be further follow-ed up for an update at a later moment in time.

Author Response

Excelent article, I applaud your work, both surgical and scientific. The entirety of the article is well written, with very clear explanation to the process, results, and important discussions. Nevertheless, I would like you to recheck the data in the tables - for example in Table 1, AFP values are very high - please check.

We would like to thank you for your compliment. The values of the tables, especially the AFP were checked and found to be correct.

The only drawback I can find to the article is the low number of cases. I hope that you will be able to conduct a larger study, maybe prospective, in the future. Also, I hope that the patients in the present study will be further follow-ed up for an update at a later moment in time.

True, the number is low and became even lower after the exclusion of the patients that are participating in the PHITT study. This database is an ongoing effort and will be followed up to evaluate our clinical outcome as well as prepare for a new publication once PHITT is concluded.